# The role of chromatin state in intron retention: A case study in leveraging large scale deep learning models

**Ahmed Daoud, Asa Ben-Hur** [ID]*

Department of Computer Science, Colorado State University, Fort Collins, Colorado, United States of America

* asa@colostate.edu

## Abstract

Complex deep learning models trained on very large datasets have become key enabling tools for current research in natural language processing and computer vision. By providing pre-trained models that can be fine-tuned for specific applications, they enable researchers to create accurate models with minimal effort and computational resources. Large scale genomics deep learning models come in two flavors: the first are large language models of DNA sequences trained in a self-supervised fashion, similar to the corresponding natural language models; the second are supervised learning models that leverage large scale genomics datasets from ENCODE and other sources. We argue that these models are the equivalent of foundation models in natural language processing in their utility, as they encode within them chromatin state in its different aspects, providing useful representations that allow quick deployment of accurate models of gene regulation. We demonstrate this premise by leveraging the recently created Sei model to develop simple, interpretable models of intron retention, and demonstrate their advantage over models based on the DNA language model DNABERT-2. Our work also demonstrates the impact of chromatin state on the regulation of intron retention. Using representations learned by Sei, our model is able to discover the involvement of transcription factors and chromatin marks in regulating intron retention, providing better accuracy than a recently published custom model developed for this purpose.

## Author summary

DNA language models have recently been presented as a methodology for quick development of accurate predictive models for biological sequence data. In this work we present an alternative approach, which is to use large scale models trained in a supervised manner over large epigenetics datasets. We demonstrate the power of this approach to create accurate and interpretable models on the problem of detecting retained introns in regions of open chromatin. Our analysis reveals a major role for transcription factors in regulating this process, made possible by the co-transcriptional nature of splicing.

**Data Availability Statement:** The code and data used for training the models and generating our results is available on GitHub at https://github.com/Addaoud/IntronRetention; a snapshot of the

repository is available on Zenodo at https://doi.org/10.5281/zenodo.13921201.

**Funding:** This material is based upon work supported by the U.S. Department of Energy, Office of Science, Biological and Environmental Research program (BER), under Award Number DE-SC-0024459. Both authors received salary under this award. The funders had no role in study design, data collection and analysis, decision to publish, or preparation of the manuscript.

**Competing interests:** The authors have declared that no competing interests exist.

## Introduction

As in natural language processing and computer vision, deep learning has changed the way we analyze large biological datasets, particularly those relating to the complexities of gene regulation in all its aspects [1]. When using deep learning to address a problem of interest you can directly train a network to solve that problem using available labeled data. The other approach, using transfer learning, is to leverage a network trained on related tasks where large training sets are available. This has become the standard approach in natural language processing and computer vision, where a standard set of models have been established as building blocks for downstream applications. Examples include extremely large NLP models such as BERT [2] and GPT-3 [3] trained in a self-supervised fashion. These are often referred to as *foundation models* [4]. The same techniques have been applied to train large transformer models on large scale protein and DNA sequence datasets, yielding models that provide useful representations of biological sequences [5–7]. However, despite their usefulness, these models have not had the level of impact of the foundation models used in NLP. The language of DNA is much noisier and difficult to interpret. And furthermore, it is incomplete: the biological function of DNA is dependent on chromatin state and the overall cellular state in which it operates. We argue that the information provided by chromatin state profiles or gene expression values is a way towards filling in this missing information and help deep learning models create more meaningful DNA sequence representations.

Prediction of chromatin state genome-wide was one of the first applications of deep learning in genomics [8, 9]. In Basset, Kelley et al. created a model to predict chromatin accessibility captured as peaks in DNase I-seq data across close to two hundred experiments [9]. DeepSEA included ChIP-seq data for transcription factor binding of 160 transcription factors across 690 datasets and 104 histone-mark profiles in addition to 125 chromatin accessibility datasets [8]. In both approaches, convolutional networks were used to model regions detected as statistically significant peaks in DNase I-seq or ChIP-seq data. These were the early supervised foundation models of chromatin state. The Basenji model [10] took the ideas from the Basset model a couple of steps further, replacing modeling of statistically significant peaks to predicting read coverage in 128-bp bins across the entire human genome. This led to a big jump in accuracy compared to the peak-based modeling employed in Basset, also the result of a shift to the use of dilated convolution, which allowed the model to learn shared patterns across larger distances. The Enformer model builds on Basenji and includes attention-based layers that are able to model long-range relationships even better than dilated convolution [11]. The Sei model is the further development of the DeepSea and DeepSea-beluga models [8, 12] and provides predictions of an extensive set of chromatin-state descriptors that include transcription factor binding, chromatin accessibility, and histone modifications for a total of 21,907 genome-wide chromatin-state profiles [13].

In this work we demonstrate the value of using chromatin state foundation models for accurate prediction of accessible regions within human introns that are subject to intron retention (IR), and demonstrate their advantage over DNA language models. It is well established that splicing is predominantly co-transcriptional [14], and that chromatin state has a strong influence on splicing outcome [15, 16]. However, while there are several hypotheses on the mechanisms of co-transcriptional regulation of splicing [15], the details of this phenomenon are still unknown. Furthermore, while many individual transcription factors are known to affect splicing outcome, the extent to which they affect splicing has yet to be determined. In earlier work we have presented a deep learning model that is capable of predicting retained introns in regions of open chromatin, and demonstrated that transcription factors have a significant impact on the regulation of this phenomenon [17]. Since the open chromatin dataset

was based on data generated by the authors of Basset, we explored models based on their architecture; in this case transfer learning had limited value, providing accuracy that is similar to that obtained by training a model from scratch. In this work we chose to use the Sei model, which is a next generation chromatin state foundation model [13]. It is a good match for the task at hand as it models a large number of characteristics of chromatin state, including transcription factor binding, histone modifications, and also uses a relatively short sequence length compared to models like the Enformer, making it easier to use. By leveraging the knowledge encoded within the pre-trained Sei model, we were able to quickly train models with higher accuracy than our earlier models; furthermore, by directly using the predicted chromatin state variables predicted by Sei, we were able to use a simple logistic regression approach that yielded a highly interpretable model based entirely on transcription factor binding predictions. While our earlier model provided evidence for the involvement of transcription factors in IR [17], the Sei-based model makes that connection more explicit, and is able to discover a much larger set of transcription factors involved in this process, several of which are already known to be associated with IR. Overall, our work demonstrates that transcription factors likely have a major role in regulation of IR, suggesting interesting directions for future work.

Our findings on the disadvantage of DNA language models are in strong agreement with recent findings in a preprint from the Koo lab [18], which came to similar conclusions using multiple genomics predictions tasks.

## Related work on splicing prediction

Over the years there have been multiple efforts to apply machine learning as a way to learn the so-called *slicing code*—a set of rules from which splice site choice can be inferred. This was done first for exon skipping [19], and more recently, for intron retention [20]. These studies used expert-determined features that were known to be associated with splicing outcome, in addition to simple sequence-based features. Later work created more sophisticated models of the splicing code that included the activity of splicing factors as measured by CLIP-seq data [21]. Similar to our work, Lee et al. [22] studied the relationship between alternative splicing and chromatin state. They represented an exon skipping event as a sequence of epigenetic signals that describe histone marks, open chromatin, DNA methylation, and nucleosome density, and trained a recurrent network to predict exon inclusion from these data. Our approach on the other hand is to start directly from sequence, and to focus on regions of chromatin within the intron rather than entire introns. This makes the problem harder, as it ignores sequence signals within splice junctions that are predictive of intron retention. However, it allows the model to focus on our objective of modeling sequence elements within regions of open chromatin that are predictive of intron retention. Finally, there have been a multiple studies that describe sequence-based deep learning methods for splice site prediction, e.g. SpliceAI [23], and tissue or condition specific splicing, usually exon inclusion, e.g. the Pangolin method [24]. Unlike our approach, these focus on modeling the splice junction region and signals adjacent to it.

## Results and discussion

In what follows we compare multiple approaches for transfer learning to capture the relation between intron retention (IR) and chromatin state, and demonstrate their merit in comparison to our earlier work [17]. To highlight the role of chromatin state in this phenomenon we use sequences that belong to regions of open chromatin as captured by DNase I-seq data, i.e. are DNAse I hypersensitive sites (DHSs). We trained models to distinguish between DHSs associated with IR from DHSs that exhibit constitutive splicing. Our data is based on the dataset created by Ullah et al. [17], which uses the genome annotations to define retained introns.

In this work we enhanced that dataset with IR events extracted from RNA-seq data (see Methods section for details). The resulting dataset contains 20,925 IR DHSs (positive examples) and 55,067 DHSs that are not known to exhibit IR (negative examples).

## Transfer learning is all you need

In order to predict intron retention, we repurposed the Sei framework by taking advantage of its trained convolutional layers. To adapt it to our problem we replaced its classification layer and fine-tuned its entire set of parameters. We refer to this model as FSei (see Fig 1). For comparison, we also trained a model using the DNABERT-2 language model as a building block [25]. The fine-tuned Sei model (FSei) outperformed the LLM-based model that uses DNA-BERT-2 (area under the precision-recall curve (AUPRC) of 0.653 compared to 0.567). The fine-tuned Sei model also performed better than a model trained from scratch that uses an architecture based on the Basenji chromatin state model [10], which had an AUPRC of 0.627 (see Fig 2 for the precision recall and ROC curves for these models). We also note that the Basenji-like model outperformed a simpler model that uses a more basic architecture similar to the Basset model [9] which achieved an AUPRC of 0.614. This underscores the effectiveness of using the Sei framework in enhancing predictive accuracy for this task, and that the chromatin representation it learns is highly relevant for the task of predicting retained introns in regions of open chromatin. Its performance advantage over the models trained from scratch likely stems from it having been trained on the entire human genome, giving it the opportunity to more fully learn the language of chromatin-based gene regulation.

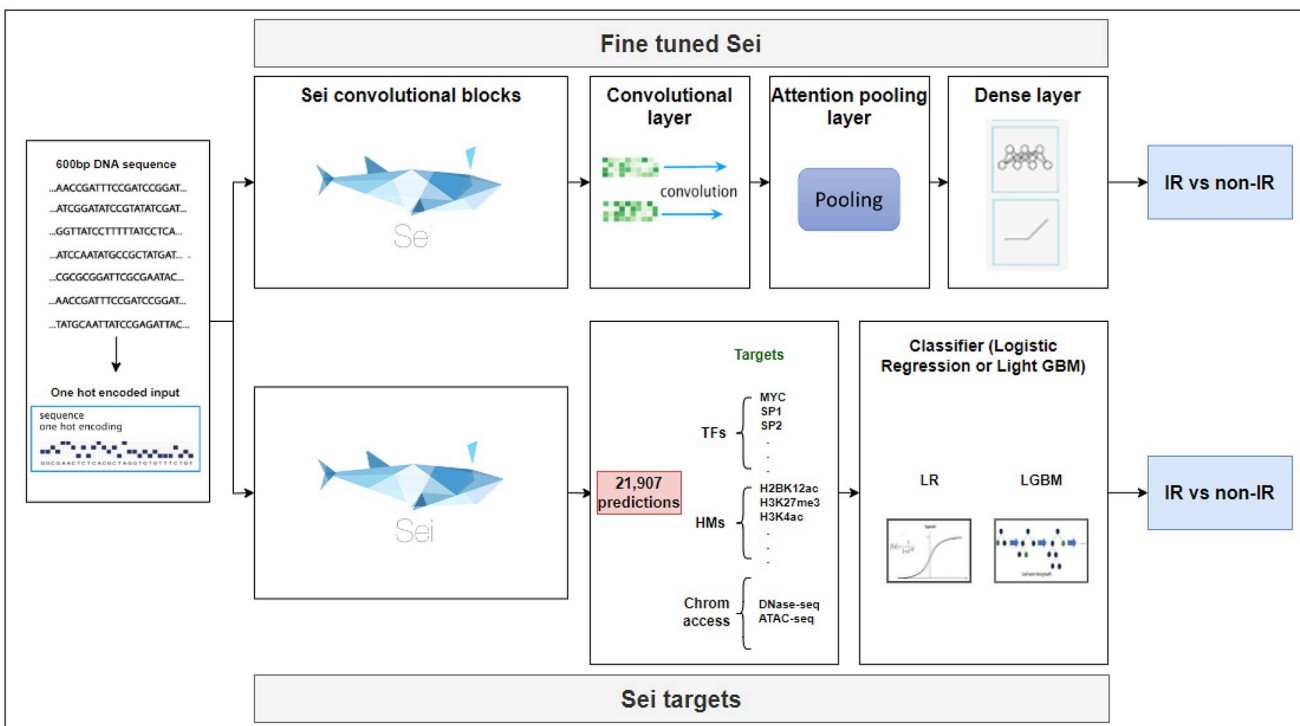

**Fig 1. Two approaches for leveraging the Sei architecture for predicting retained introns.** Top: The Fine-tuned Sei version uses the representation learned by the Sei convolutional layers, adding an additional (optional) convolutional block plus pooling and fully connected and output layers. Bottom: The Sei-outputs version uses the chromatin targets learned by Sei (transcription factor binding, histone modifications, and chromatin accessibility) and uses these as input to a logistic regression or LightGBM classifier.

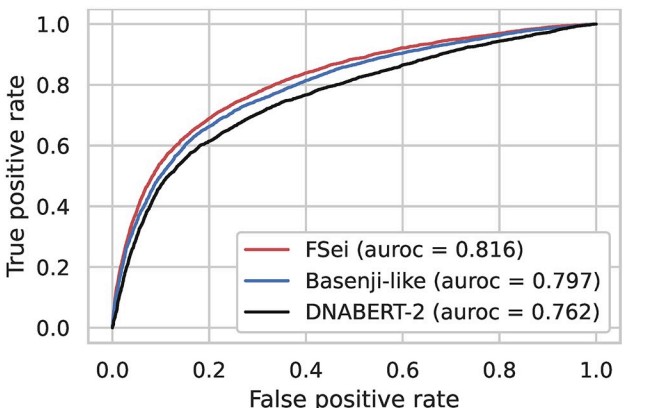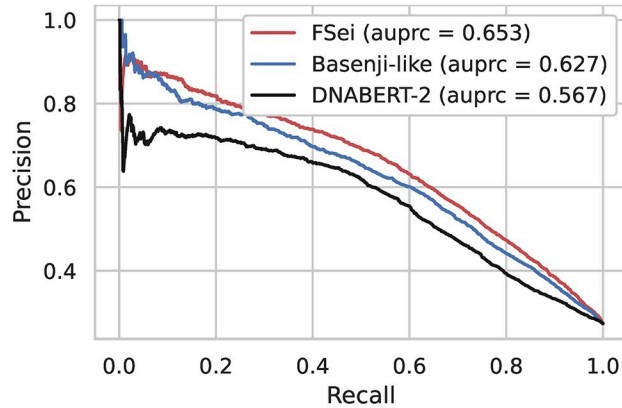

**Fig 2. Classification accuracy of deep learning models for prediction of intron retention.** We compare the performance of the fine-tuned Sei model (FSei) with a model that uses embeddings generated by DNABERT-2 and a Basenji-like architecture trained from scratch. We provide the ROC and precision-recall curves along with the area under the ROC curve (auroc) and area under the precision recall curve (auprc) of each model in the legend.

Next, we explored a more sophisticated way of leveraging the Sei embeddings by adding an additional convolutional layer, followed by an attention pooling layer, a dense layer, and an output layer as shown in Fig 1. This more complex way of using Sei embeddings provided a slight improvement in accuracy, increasing the AUPRC from 0.653 to 0.656. The same fine-tuning block did not provide any improvement in the performance of DNABERT-2 (see Table 1). Also noteworthy, is that freezing the weights of the Sei model provided lower accuracy (see Table 1). Our results indicate that using a complex fine-tuning block is not necessary in our case with our input data being sequences of length 600bp; however, for classification problems on longer sequences, we expect this approach to provide greater benefit.

## Interpretable prediction of IR using Sei-targets

While the fine-tuned Sei approach described above provides high accuracy, deriving biological insight from it can be challenging. Therefore, we provide an alternative, much simpler approach that is easily interpretable, faster to train, and achieves prediction accuracy that is close to the fine-tuned Sei model. The alternative approach, which we refer to as *Sei-targets*,

**Table 1. Classification accuracy.** We compare the accuracy of the following methods: FSei—fine tuned Sei model that simply replaces the classification head of the original model; FSei* adds a fine-tuning block that includes a convolutional layer; FSei-frozen has the same architecture as FSei, except that all layers except for the classification block are held fixed; the two Sei-targets models trained a logistic regression or LightGBM classifier based on Sei predictions; the DNABERT-2 model uses DNABERT-2 embeddings, while DNABERT-2* uses the same fine-tuning block used in FSei*. The two trained-from-scratch models are based on the architectures used in Basenji [10] and Basset [9]. The accuracy of the two best performing models is highlighted in boldface.

| Model type | Model | AUROC | AUPRC |
|---|---|---|---|
| Sei embeddings | FSei | **0.816** | **0.653** |
| | FSei* | **0.817** | **0.656** |
| | FSei-frozen | 0.813 | 0.648 |
| Sei targets | Logistic regression | 0.807 | 0.636 |
| | LightGBM | 0.798 | 0.624 |
| Language model | DNABERT-2 | 0.762 | 0.567 |
| | DNABERT-2* | 0.751 | 0.554 |
| Trained from scratch | Basenji-like | 0.797 | 0.627 |
| | Basset-like | 0.795 | 0.614 |

uses the outputs produced by Sei, which are 21,907 chromatin-state profiles that describe transcription factor binding, open chromatin, and chromatin modifications. These outputs are provided as input features to a simple classifier, either logistic regression or LightGBM trained to predict retained introns based on these 21,907 descriptors of chromatin state provided by Sei (see Fig 1).

The Sei-targets model comes in three flavors depending on which subset of Sei targets were provided as input:

1. All targets: this model uses all 21,907 Sei chromatin profiles as input.

2. TF model: this model uses the 9,471 predictions associated with transcription factors (TFs) and other DNA binding proteins were kept when creating this dataset. This dataset includes around 1,000 non-histone DNA-binding proteins.

3. HM model: this model uses the 10,064 Sei outputs associated with histone marks. This includes 77 different histone targets and marks.

Based on these three sets of targets we built three logistic regression models to predict intron retention using these sets of targets as input. We observed that all three logistic regression models had very similar performance with AUPRCs between 0.636 and 0.641 (see Fig 3). The TF-targets model outperformed the Bassenji-like model (AUPRC of 0.627 compared to 0.636), and performed only slightly worse than the fine-tuned Sei model. For further comparison, we also trained a LightGBM model based on the TF-targets data, and its performance was similar to the logistic regression model (see Fig 4). Thus, we conclude that predicted chromatin-state profiles from the Sei framework in conjunction with a simple logistic regression approach are capable of predicting DHSs associated with retained introns with high accuracy.

Our observation on the redundancy between histone marks and TF binding as predictive variables has also been observed in the context of prediction of gene expression [26, 27]. This is not surprising, since histone marks contribute to TF binding [28–30], and vice versa—TFs help recruit histone modifiers [29–31]. In this work we chose to model IR-associated open chromatin directly from sequence. An alternative is to use epigenetic data, represented as a sequence of histone modifications and other chromatin state variables using experimental data rather than predicting them from sequence. This was the modeling choice made e.g. by Lee et. al [22] for predicting exon skipping. Direct modeling from sequence has several advantages.

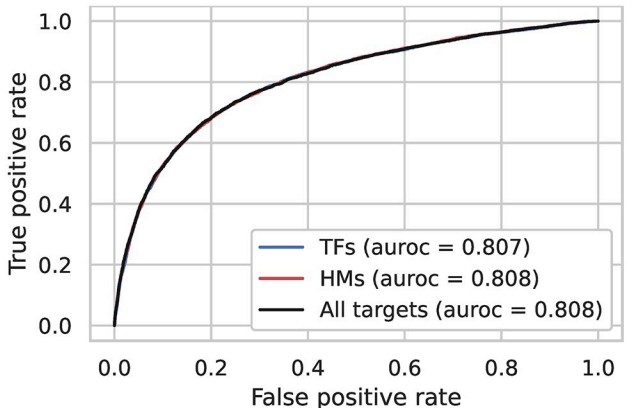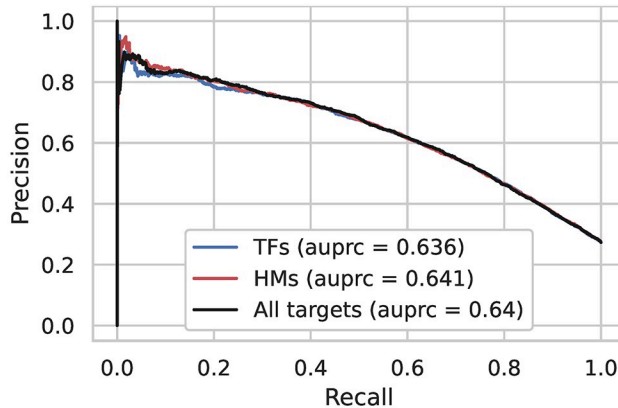

**Fig 3. Classification accuracy of the logistic regression model using the three different datasets constructed from Sei predictions.** We provide the ROC and precision-recall curves along with the auroc and auprc of each dataset in the legend.

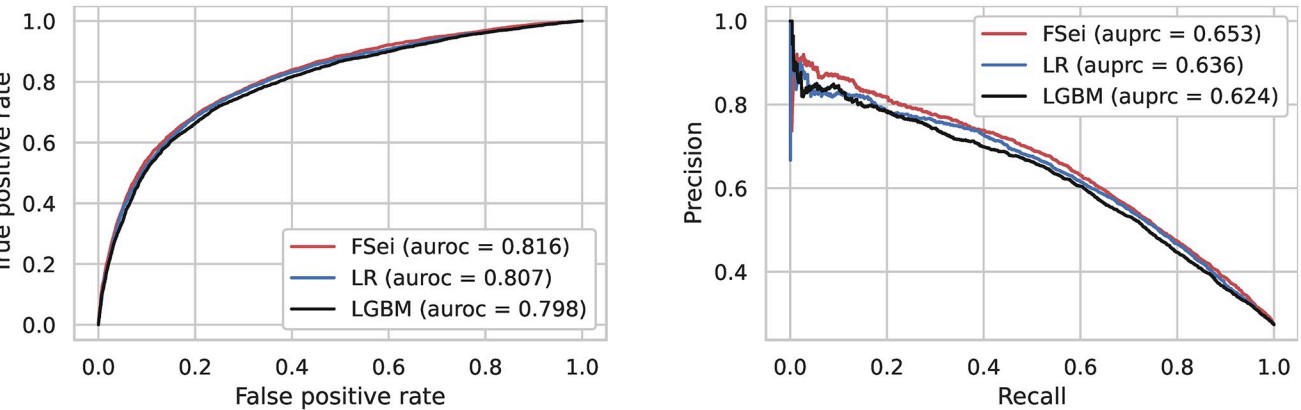

**Fig 4. Classification accuracy of deep learning models for prediction of intron retention.** We compare the performance of fine-tuned Sei (FSei), and two models that use the TF outputs generated by Sei as features: a simple logistic regression model (LR) and LightGBM (LGBM). We provide the ROC and precision-recall curves along with the auroc and auprc of each model in the legend.

First, it allows for discovery of sequence elements that drive alternative splicing. Second, it does not require the assembly and processing of large genome-wide experimental datasets. Instead, our approach relies on the heavy lifting being done in assembly of the large scale data-sets sets and training of a chromatin state model such as Sei.

## TF involvement in IR

An important advantage of the logistic regression approach over the fine-tuned Sei model is that it is amenable to easy interpretation by looking at the weights assigned to the different tar-gets. To quantify the contribution and involvement of TFs in IR, we considered the weights of the logistic regression model and the feature importance assigned by the LightGBM classifier (see Fig 5). Before discussing the rankings, we note that a given DNA binding protein can occur multiple times on the list, as a given protein might have been assayed in different cell lines or tissues and therefore associated with multiple Sei targets. And indeed, among the

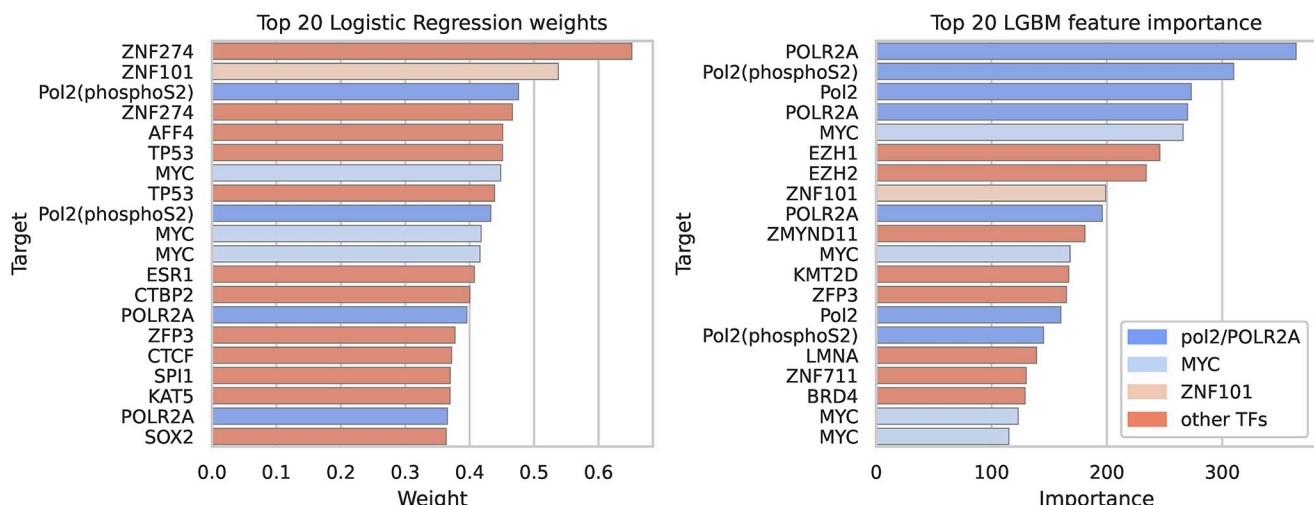

**Fig 5. The top 20 targets that contribute to IR in logistic regression (left) and LightGBM (right).** Targets are identified by their color.

9,471 DNA-binding targets there are only about 1,100 unique proteins. As a result, a given DNA binding protein will be assigned multiple ranks, reflecting the different activities or levels of activity that a protein can have in different tissues or conditions.

We observe many similarities in the rankings produced by the two classifiers, and are able to provide evidence for the involvement of several of the top ranking targets in alternative splicing. First, both classifiers ranked Pol II and modified versions of it (e.g. Pol II modified by phosphorylation of Serine 2) near the top of the list. It is well known that Pol II elongation rate is coupled with alternative splicing, and that modifications of its carboxy terminal domain (CTD) also affect splicing outcome [32]. Therefore it's also not surprising that POLR2A, a major component of Pol II, also appears high on both lists. AFF4 is a core component of the "super elongation complex", and affects the rate of Pol II elongation and hence alternative splicing [33]. MYC is one of the top ranking TFs in both lists. It is a well established oncogene that is dysregulated in many cancers, and several studies have linked it to alternative splicing in the context of cancer [34–36]. We also note that cancer is characterized by widespread intron retention even in the absence of spliceosomal mutations [37], and we hypothesize that this may be partly as a result of changes in chromatin state. CTCF is an extremely well studied DNA-binding protein, and is in the top 20 list for logistic regression. It is known to regulate alternative splicing through multiple mechanisms related to its ability to remodel chromatin and affect Pol II elongation [38]. BRD4 was recently shown to regulate alternative splicing [39]. It was ranked 49 by the logistic regression model and 18 by the LGBM classifier.

We observed that some TFs have both large positive weights and large negative weights, indicating that a given TF can both promote IR and constitutive splicing in different cellular contexts. The top 100 and bottom 100 ranked TFs had an intersection of 22 TFs that include CTCF, BRD4, POLR2A, and MYC. As mentioned above, this is possible since a given TF can be represented in multiple Sei targets describing its activity in different tissues or cell lines.

In our earlier work we have identified multiple TFs as associated with IR, and provided evidence from ChIP-seq experiments for the association of EGR1, ZNF263, SP1, SP2, MAZ, and ZBTB7A [17]. The first three were in the top 5% of targets identified by logistic regression while ZBTB7A, MAZ, and SP2 are somewhat lower and had ranks of 1,150, 1,600, 2,229 out of 9,741, respectively. Overall, in the top 5% we have identified 215 different DNA binding proteins with high logistic regression weights. The complete list of ranked Sei targets associated with IR is provided in the github pages of this project.

## How do TFs regulate IR?

The results of the Sei-targets model based on TFs suggest that regulation of IR is coordinated by a large number of TFs. While the literature contains ample discussion of the connection of the chromatin landscape and splicing [16], there are only isolated examples of TFs associated with IR, and their role as a whole has not been widely explored. There are several ways in which chromatin state has been proposed to affect splicing, and TFs fit well within these mechanisms. In the *kinetic model*, the speed of Pol II elongation modulates the availability of splice sites thereby affecting splice site choice [16]. TFs, through their effect on chromatin architecture, are capable of orchestrating elongation rates either directly, or through their effect on histone modifications and other factors that affect nucleosome positioning. An alternative way in which TFs can affect splice site choice is through the *recruitment model*: We propose that like the CTD domain of Pol II [16], TFs can interact with splicing factors and function to orchestrate their co-transcriptional recruitment. Further experimental work is required to explore these hypotheses.

### Interpreting the fined-tuned Sei model

While more difficult, it is also possible to interpret the fine-tuned Sei model and obtain associations of TFs with IR. Using the methodology described in the Methods section, which uses positions identified using the Integrated Gradients method, we obtained a ranking of 949 human TFs based on their preference for occurrence in retained introns compared to non-retained introns. The TFs EGR1, SP1, ZBTB7A, and ZNF263 identified by Ullah et al. [17] were all in top 50, while MAZ and SP2 was ranked 124 and 93 respectively. The ranking produced by the Sei-targets approach is not directly comparable to the one produced by Integrated Gradients since each of them is based on a very different set of TFs. Among the two lists (1,097 unique TFs in Sei-targets, and 949 unique TFs in Jaspar), only 334 appeared in both. Thus, the two rankings provide complementary information on the involvement of TFs in IR. We also identified TFs that were strongly associated with non-retained introns, and those likely have a role in constitutive splicing. A complete list is found on the github page of the project. Finally, we note that the Integrated Gradients approach we described can potentially be used to interpret the LLM-based model, although its use of non-overlapping k-mers in creating its embeddings could make this more of a challenge.

## Conclusions

This study showed the effectiveness of using Sei, a foundation chromatin model, for the downstream application of predicting intron retention, which is known to be co-transcriptionally regulated, and hence impacted by chromatin state. The high accuracy of this approach indicates the large contribution of chromatin state to this phenomenon. The pre-trained model produced superior results compared to building a model from scratch, and also improved on a model based on the DNA language model DNABERT-2. This can be understood from the fact that the Sei model captures more of the complexities of chromatin state due to the much larger datasets used to train it: it was trained on the entire human genome across over twenty thousand experimental genome-wide chromatin-state profiles. Furthermore, the model based on Sei targets was also easier to interpret, enabling us to find biologically meaningful associations between TFs and other DNA binding proteins and splicing regulation. Our findings are in strong agreement with recent work which came to similar conclusions using multiple genomics predictions tasks [18].

Our results showed that a large number of transcription factors are likely involved in intron retention. However, the mechanism by which this layer of regulation occurs remains to be established. TFs can affect splicing outcome in various ways: either by establishing chromatin state, which in turn affects splicing, or more directly by recruiting splicing factors, and further experiments are required to elucidate the molecular mechanisms of TF regulation of splicing.

## Methods

### Data

Our models were trained using intronic DHSs labeled as IR (positive examples) or non-IR (negative examples) depending on whether or not the DHS is within an intron that exhibits IR. While in our our original publication [17] we only used IR events extracted from gene models, in this work we extended the data with the addition of IR events detected in RNA-seq. To prepare the data, we used a set of DNase I hypersensitive sites identified in a large collection of DNase I-seq datasets analyzed by Kelley et al. [9]. We then used SpliceGrapher [40] to extract IR events from the Ensembl GRCh37 (hg19) reference annotations. DHSs, represented as sequences of length 600bp, that overlap retained introns constitute the positive samples;

**Table 2. Dataset summary.** The first row describes the number of IR and non-IR events overlapping DHSs extracted from the gene annotations. The next row shows the number of additional IR events, and the reduction in the number of non-IR events. We also show the total number of events, and the total number of training examples as a result of data augmentation by reverse complementation.

|  | IR | non-IR |
|---|---:|---:|
| Events from gene annotations | 16,231 | 55,193 |
| Events from RNA-seq | 4,694 | -126 |
| Total events | 20,925 | 55,067 |
| After reverse complementation | 41,850 | 110,134 |

intragenic DHSs that did not overlap an IR event were labeled as negative examples. We then used CD-hit [41] at a similarity threshold of 80% to remove redundancy between the test sets and the training and validation sets.

To obtain a larger set of positive examples, we extracted additional IR events from RNA-seq data in the K562 cell line (GEO Accession: GSM958731); the reads were aligned to the genome with STAR [42] with its default parameters, followed by processing with SpliceGrapher [40] to obtain the list of retained introns. We then compared the DHSs in the original dataset with this list and found 126 negative examples (labeled as non-IR) overlapping by at least 500 bps with a retained intron. We relabeled those samples as IR. We also found 4,568 new DHSs, not present in the original dataset, overlapping by at least 500 bps with a retained intron and included those as well.

Finally, we used reverse complement for data augmentation. The final dataset contains 151,984 DNA sequences: 41,850 DNA sequences associated with IR and 110,134 labeled as non IR (see Table 2). To prepare the sequences for use with deep learning, we used one-hot encoding. In one-hot encoding, a sequence is represented as a $4 \times N$ matrix where $N$ is the DNA sequence length. The columns of the matrix represent each position in the sequence assigning a non-zero value at a place corresponding to one of the four DNA nucleotides. The resulting dataset is available in the GitHub repository at `data/final_data.csv`; reverse complementation is performed while the data is loaded.

## The Sei model

Sei is a deep learning model that predicts the chromatin state of a DNA sequence. It is trained on the entire human genome across many experimental modalities to predict 21,907 genome-wide *cis*-regulatory targets across more than 1,300 cell lines and tissues, including 9,471 TF binding profiles, 10,064 histone mark profiles and 2,372 chromatin accessibility profiles [13]. We split the Sei targets according to the following grouping:

- Targets characterizing chromatin accessibility using DNase I-seq and ATAC-seq across different cell lines and tissues, totalling 2,372 targets.

- Targets associated with histone marks. (10,062 targets).

- The rest of the targets, totalling 9,471 are nonhistone DNA-binding proteins. For simplicity, we refer to them as transcription factors, which constitute the majority.

Sei was trained to predict these profiles from a 4kb sequence represented using one-hot-encoding. The model architecture is composed of three parts: (1) convolutional blocks with dual linear and nonlinear paths; (2) residual dilated convolution layers; and (3) a pooling and classification block. The classification block is one dense layer followed by an output layer of size 21,907.

### Fine-tuned Sei for predicting IR

This model uses the pre-trained Sei model, repurposing it to predict intron retention. We compare two approaches for using Sei embeddings. The first, simply replaces the classification layer of Sei with a binary classification layer. In the other approach we take the pre-trained Sei convolutional layers up to the B-spline layer. We then apply an additional convolutional layer, followed by the same attention pooling strategy used in the Enformer model [11]. This is followed by one dense layer and the output layer.

### The Basset-like model

This model has three convolutional layers followed by three fully connected layers constructed using the same parameters used in [17]. It also uses BatchNorm layer [43] and max-pooling layers to reduce internal covariate shifts and improve invariance to small shifts in the input sequence.

### The Basenji-like model

This model, a simplified version of the Basenji2 [44] model, has one linear convolutional layer and six blocks of linear and non-linear convolutional layers, followed by three blocks of dilated convolutional layers and the same block used to fine-tune the Sei model. We also used batch-norm layers, GELU as the activation function following [44], and skip connections around the dilated convolutional layers.

### DNABERT-2 for predicting IR

DNABERT-2 is an improved version of the DNA large language model DNABERT, that makes use of an efficient tokenizer and a variety of techniques to get around limits on input length, save time and memory use, with enhanced model performance [7]. DNABERT-2 is a large language model that uses a Transformer Encoder architecture similar to BERT [2]. The authors trained DNABERT-2 with the masked language modeling loss with a mask ratio of 15% on a multi-species genome dataset. The model can be loaded using the Transformers library HuggingFace (Wolf et al. [45]). To leverage DNABERT-2 to predict retained introns we used their BertForSequenceClassification model, available on HuggingFace, which uses the pooled embeddings produced by DNABERT-2 followed by a prediction layer. As with Sei, we explored the use of a more sophisticated way of using the embeddings using the same fine-tuning block as for the Sei model.

### Logistic regression

We used PyTorch to implement the logistic regression model and trained it to predict IR on the basis of Sei targets as input. Additionally, since the data is imbalanced, we used an imbalanced sampler when training the model.

### Light Gradient Boosting Machine

LightGBM [46] is a state-of-the-art gradient boosting model with decision trees as the base learning algorithm. It offers the ability to evaluate variable importance, similarly to random forests. Like the logistic regression model, the input to this classifier is the set of Sei targets. In this work we used version 4.0.0 provided by Microsoft, with GPU-acceleration support. We ran the model with the following parameters: n_estimators = 1000 which is the number of trees, max_depth = −1, num_leaves = 50, bagging_freq = 1, is_unbalance = *True* which sets the imbalanced mode on, since the data exhibits clear imbalance.

**Table 3. Model hyper-parameters and training statistics.** Maximum epochs is the maximum number of epochs allowed for training; after not showing performance improvement for "Early stopping epochs" model training is stopped; the actual number of epochs used is provided in "Training epochs".

| Model | Parameter | Maximum epochs | Early stopping epochs | Actual training epochs | Training time (min) |
|---|---|---|---|---|---|
| Fine-tuned Sei | | 100 | 10 | 20 | 114 |
| DNABERT-2 | | 100 | 10 | 18 | 323 |
| Basset-like | | 100 | 10 | 19 | 2 |
| Basenji-like | | 100 | 10 | 13 | 23 |
| LR | | 1000 | 50 | 190 | 12 |
| LGBM | | 1000 | 50 | 230 | 12 |

### Training, validation, and testing

The data was split into training, validation, and test sets containing 80%, 10%, and 10% of the data, respectively. For all models, we employed a GPU-accelerated training approach. To prevent overfitting and optimize model performance, we implemented early stopping based on the improvement of the validation loss over a predetermined number of consecutive epochs (the parameter "early stopping epochs" in Table 3). Additionally, to manage computational resources, we set a predefined maximum number of epochs for each model. All the models were trained with the binary crossentropy loss function. The Sei and DNABERT-2 models were fine-tuned using SGD with a large value of momentum (0.95), while all other models were trained using the AdamW optimizer (Adam with weight decay) with the default weight decay parameter. The details of the hyper-parameter space is summarized in Table 3. We also used a scheduler that linearly increases the learning rate, for warmup_steps, starting from warmup_begin_lr until it reaches max_lr, and then decreases it to final_lr following a cosine scheduler [47]. The scheduler's parameters were chosen based on an exploratory analysis on the behaviour of the training and validation losses for each model separately. All experiments were performed on a machine with an NVIDIA GeForce GTX TITAN X GPU with 12GB of memory.

### Attribution with Integrated Gradients

We propose an approach to score the relevance of each TF using Integrated Gradients (IG) [48], which is an efficient attribution method that assigns an importance score to each position in the sequence. In the first step, we selected all the DNA sequences in the validation and test datasets that were correctly predicted by the fine-tuned Sei with a probability above 0.7. This reduced the number of sequences to 1178 IR sequences and 16631 non-IR sequences. Then, we applied Integrated Gradients on the one hot encoded selected sequences to extract all hot spots (positions) that have a gradient value higher than 70% of the maximum value in the same sequence. We used the IntegratedGradients function provided by captum, a model interpretability library for PyTorch [49], to compute the scores. A total of 109,974 hot spots were selected in non-IR sequences and 5,521 in IR sequences. To map the resulting IG scores to TF binding enrichment in IR vs non-IR sequences, we used the Jaspar database [50] to obtain motifs of known TFs. We scored each IG hot spot against the set of Jaspar motifs and kept only those that had a TF score of at least 80% of the maximum score for its motif. For each TF we then scored its enrichment in IR by subtracting the fraction of IR IG hot spots that have a given motif and the fraction of non-IR IG hot spots that have the motif.

### Acknowledgments

We thank Saira Jabeen for her help with the RNA-seq data.

## Author Contributions

**Conceptualization:** Ahmed Daoud, Asa Ben-Hur.

**Formal analysis:** Ahmed Daoud, Asa Ben-Hur.

**Funding acquisition:** Asa Ben-Hur.

**Investigation:** Ahmed Daoud, Asa Ben-Hur.

**Methodology:** Ahmed Daoud, Asa Ben-Hur.

**Project administration:** Asa Ben-Hur.

**Resources:** Asa Ben-Hur.

**Software:** Ahmed Daoud.

**Supervision:** Asa Ben-Hur.

**Visualization:** Ahmed Daoud.

**Writing – original draft:** Ahmed Daoud.

**Writing – review & editing:** Asa Ben-Hur.

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
