## [Decision Letter · Decision Letter 0]

9 Sep 2024

Dear Dr. Ben-Hur,

Thank you very much for submitting your manuscript "The role of chromatin state in intron retention:  a case study in leveraging large scale deep learning models" for consideration at PLOS Computational Biology. As with all papers reviewed by the journal, your manuscript was reviewed by members of the editorial board and by several independent reviewers. The reviewers appreciated the attention to an important topic. Based on the reviews, we are likely to accept this manuscript for publication, providing that you modify the manuscript according to the review recommendations.

Reviewer 1 had difficulties installing the repository, and the results using Fsei did not match the results reported in the paper. Your revision should address these comments (as well as the other comments from both reviewers).

Sincerely,

Shihua Zhang

Section Editor

PLOS Computational Biology

Shihua Zhang

Section Editor

PLOS Computational Biology

Reviewer 1 had difficulties installing the repository, and the results using Fsei did not match the results reported in the paper. Your revision should address these comments (as well as the other comments from both reviewers).

Reviewer's Responses to Questions

**Comments to the Authors:**

Reviewer #1: Overall impressions:

---

The paper’s argument for the use of foundation models in genomic contexts because of the complexity of the genome is interesting. The authors make the case specifically for leveraging a chromatin state foundation model for prediction of intron retention and compare its merits against other deep learning models.

- Considering that the paper is framed as a case study, I think it would be enhanced by further discussion of the biological motivation for the work.

- In Section 2, the paper would benefit from further clarification/discussion of the occurrence of duplicates in the ranked list of DNA binding proteins and how that affects the analysis of results.

Code review:

---

The code located at the corresponding GitHub repository does not completely work off the shelf, and I was not able to fully reproduce the results in the manuscript.

Reproducibility:

1. I was unable to produce results for either the LR, LGBM, or FDNABert models using the code and instructions provided. See errors list below for details.

2. Running the ConvNet (called Bassenji-like in the manuscript?) and Basset models with the instructions provided on GitHub reproduced the AUROC and AUPRC results to within around 0.001-0.002 of the numbers reported in the paper/the authors' supplementary figures on GitHub.

3. There is an "FDNABert" and a "DNABert" model available on GitHub but only one DNABert model, I believe, mentioned in the paper.

4. Running FSei with the instructions provided on GitHub did not reproduce the results demonstrated in the paper. In the paper, FSei comes out clearly on top of the pack, with AUROC of 0.815 and AUPRC 0.654. In my run of the code, FSei reported an AUROC of 0.786 and an AUPRC of 0.602. These metrics are below those for Basset and ConvNet/Bassenji. See my results reported below:

- FSei | AUROC: 0.786 | AUPRC: 0.602

- ConvNet | AUROC: 0.799 | AUPRC: 0.624

- Basset | AUROC: 0.796 | AUPRC: 0.616

- DNABert (not FDNABert) | AUROC: 0.754 | AUPRC: 0.562

Errors:

A possibly non-exhaustive list of issues is below; I recommend the code be tested in a fresh environment and the repository updated from there.

1. Only the pip install line for requirements.txt runs (not the conda install line).

2. requirements.txt is missing biopython, transformers, and prettytable packages.

3. “sei.pth” also needs to be copied from the Sei resources; this is not part of the setup instructions.

4. LR.py exits with an sklearn error that originates on LR.py line 127:

“[…]/sklearn/metrics/_ranking.py", line 731:

in _binary_clf_curve

raise ValueError("{0} format is not supported".format(y_type))

ValueError: unknown format is not supported

5. Installing the packages in requirements.txt loads a version of lightgbm (4.0.0) that is not compatible with the code in LGBM.py. LGBM.py passes the argument “verbose_eval” to lgb.cv(); this argument to cv() was removed after lightgbm 3.3.5.

6. After rolling the lightgbm version back to 3.3.5 to address issue #5 above, LGBM.py still exits with an error:

“LGBM.py line 62, in lgb_eval:

return (np.array(cv_result["auc"])).max() :

KeyError: 'auc'”

7. FDNABert.py exits with an error because it cannot import "train_model" from src.train_utils.py. There is no train_model function in train_utils.py.

Organizational notes:

1. Setup of the pretrained Sei model is a bit unclear, as the user has to comb through the Sei repository to find the pieces of those authors’ instructions that are relevant. Consider including all relevant instructions in the body of the IntronRetention Readme.

2. It would be convenient for the user to be able to replicate the results from the paper simply by following the instructions on GitHub. There are a few instances where additional steps are needed. For instance, the .json files for the LR and LGBM models currently point to an output directory that won’t exist unless the user knows to create it when running Preprocess_data_to_numpy.py, and that is not mentioned in the preprocessing instructions.

3. It would be helpful for the names of the .py files to match the names of the models in the paper (for instance, in the case of the Bassenji-like model).

4. I believe that “IntronRetention/data/final_data.csv” is the dataset referenced in Section 4, Table 1. Consider clarifying this in Section 4.

Reviewer #2: - Abstract typo - "DNA langauage model DNABERT-2" line 21

- Well written introduction describing current state but deep learning approaches for intron retention prediction have not been mentioned. I do see a few when I search for it. Even if not relevant to current method, might be worth introducing it and comparing and contrasting.

-line 106 - Sound choice of using SEI framework but perhaps an explanation of the choice of another convolutional layer with pooling and dense layer being added could have been added. In the methods section a mention is made about it being like Enformer model but that uses transformer blocks as well. (7 convolutional blocks with pooling, 11 transformer blocks, and a cropping layer)

**Have the authors made all data and (if applicable) computational code underlying the findings in their manuscript fully available?**

Reviewer #1: Yes

Reviewer #2: Yes

PLOS authors have the option to publish the peer review history of their article (what does this mean?). If published, this will include your full peer review and any attached files.

Reviewer #1: No

Reviewer #2: No

Figure Files:

Data Requirements:

Reproducibility:

References:

---

## [Decision Letter · Decision Letter 1]

30 Dec 2024

Dear Dr. Ben-Hur,

Happy Holidays! We are pleased to inform you that your manuscript 'The role of chromatin state in intron retention:  a case study in leveraging large scale deep learning models' has been provisionally accepted for publication in PLOS Computational Biology.

We thank you for your careful consideration of reviewers' comments and your attention to technical requests regarding your software repository. I (JSB) personally appreciate that software repositories for research are a moving target, and keeping the repository synchronized with a manuscript under review can be hectic. From the reviews, I am satisfied that all the problems have been resolved except for one, a problem with installing the LightGBM method. My decision is to accept because LightGBM does not perform as well as Logistic Regression, and in that sense the results aren't necessary for your paper (although they do strengthen your paper). I also note that LightGBM might be a dependency from a different repository, and fixing problems in someone else's repo is beyond a reasonable scope.

Best regards,

Joel S. Bader, PhD

Guest Editor

PLOS Computational Biology

Shihua Zhang

Section Editor

PLOS Computational Biology

Reviewer's Responses to Questions

**Comments to the Authors:**

Reviewer #1: Updated Manuscript:

- The additional biological context throughout is useful.

- The addition of Table 1 is a helpful summary of results.

Updated Code/GitHut Repo:

1. Install and setup are now much smoother!

2. FSei AuROC and AuPRC now closely aligns with reported results: 0.81533... (mine) vs 0.816 (reported) and 0.65355... (mine) vs 0.653 (reported).

3. FSei-frozen runs and agrees with reported results.

4. LR model now runs and agrees with reported results.

5. Instructions/parameters for running DNABERT2*/FDNABert and FSei* to reproduce Table 1 appear to be missing from the README. It would be helpful to have these steps explicitly given.

6. Running the LGBM model still produces an error. I have Python 3.10, numpy 2.1.1, and lightgbm installed as described in the README and am seeing the following error: ".../python3.10/site-packages/lightgbm/basic.py", line 299, in _list_to_1d_numpy

return np.array(data, dtype=dtype, copy=False)ValueError: Unable to avoid copy while creating an array as requested.

If using `np.array(obj, copy=False)` replace it with `np.asarray(obj)` to allow a copy when needed (no behavior change in NumPy 1.x)."

Reviewer #2: Looks good. My concerns have been addressed.

**Have the authors made all data and (if applicable) computational code underlying the findings in their manuscript fully available?**

Reviewer #1: Yes

Reviewer #2: Yes

PLOS authors have the option to publish the peer review history of their article (what does this mean?). If published, this will include your full peer review and any attached files.

Reviewer #1: No

Reviewer #2: No

---

## [Editor Report · Acceptance letter]

4 Jan 2025

PCOMPBIOL-D-24-01050R1 

The role of chromatin state in intron retention:  a case study in leveraging large scale deep learning models

Dear Dr Ben-Hur,

I am pleased to inform you that your manuscript has been formally accepted for publication in PLOS Computational Biology. Your manuscript is now with our production department and you will be notified of the publication date in due course.

With kind regards,

Zsofia Freund
